# Insomnia Associated with Tinnitus and Gender Differences

**DOI:** 10.3390/ijerph18063209

**Published:** 2021-03-19

**Authors:** Kneginja Richter, Melanie Zimni, Iva Tomova, Lukas Retzer, Joachim Höfig, Stefanie Kellner, Carla Fries, Karina Bernstein, Wolfgang Hitzl, Thomas Hillemacher, Lence Miloseva, Jens Acker

**Affiliations:** 1Outpatient Clinic for Sleep Disorders and Tinnitus, University Clinic for Psychiatry and Psychotherapy, Paracelsus Medical University, 90419 Nuremberg, Germany; Melanie.Zimni@klinikum-nuernberg.de (M.Z.); iva.tomova@klinikum-nuernberg.de (I.T.); lukas.retzer@th-nuernberg.de (L.R.); joachim.hoefig@klinikum-nuernberg.de (J.H.); carla.fries@gmx.de (C.F.); karina.bernstein@fau.de (K.B.); Thomas.Hillemacher@klinikum-nuernberg.de (T.H.); 2Faculty for Social Work, Technical University of Applied Sciences Nuremberg Georg Simon Ohm, 90489 Nuremberg, Germany; stefanie.kellner@stud.uni-bamberg.de; 3Faculty for Medical Sciences, University Goce Delcev Stip, 2000 Stip, North Macedonia; lmiloseva@gmail.com; 4Research Office (Biostatistics), Paracelsus Medical University, 5020 Salzburg, Austria; wolfgang.hitzl@pmu.ac.at; 5Department of Ophthalmology and Optometry, Paracelsus Medical University, 5020 Salzburg, Austria; 6Research Program Experimental Ophthalmology and Glaucoma Research, Paracelsus Medical University, 5020 Salzburg, Austria; 7KSM Clinic for Sleep Medicine Bad Zurzach, 5330 Bad Zurzach, Switzerland; jens.acker@gmx.ch

**Keywords:** tinnitus, insomnia, sleep, sleep disturbance, depression, gender difference

## Abstract

Chronic tinnitus causes a decrease in well-being and can negatively affect sleep quality. It has further been indicated that there are clinically relevant gender differences, which may also have an impact on sleep quality. By conducting a retrospective and explorative data analysis for differences in patients with tinnitus and patients diagnosed with tinnitus and insomnia, hypothesized differences were explored in the summed test scores and on item-level of the validated psychometric instruments. A cross-sectional study was conducted collecting data from a sample of tinnitus patients (*n* = 76). Insomnia was diagnosed in 49 patients. Gender differences were found on aggregated test scores of the MADRS and BDI with men scoring higher than women, indicating higher depressive symptoms in men. Women stated to suffer more from headaches (*p* < 0.003), neck pain (*p* < 0.006) and nervousness as well as restlessness (*p* < 0.02). Women also reported an increase in tinnitus loudness in response to stress compared to men (*p* < 0.03). Male individuals with tinnitus and insomnia have higher depression scores and more clinically relevant depressive symptoms than women, who suffer more from psychosomatic symptoms. The results indicate a need for a targeted therapy of depressive symptoms in male patients and targeted treatment of psychosomatic symptoms, stress-related worsening of insomnia and tinnitus in women.

## 1. Introduction

Insomnia and chronic tinnitus can cause a considerable amount of suffering in everyday life for those affected. The prevalence of chronic insomnia has been reported to range between 10 and 80% with an average of 40% [1], with prevalence rates typically being higher in women than men [2]. The prevalence of tinnitus has been estimated between 10 and 15% in the adult population. From those, 20% will require clinical intervention and 80% are not particularly bothered by tinnitus [3,4]. Previous studies have shown that chronic tinnitus may be accompanied by comorbidities such as anxiety, depression and insomnia, and can have a negative impact on sleep quality, in a sense that the increased severity of sleep disruption is associated with increased tinnitus severity [5,6,7,8,9,10]. In patient groups with tinnitus and chronic insomnia, high emotional distress is associated with the patient’s tinnitus [11,12]. Ultimately, the overall distress affects every individual experiencing insomnia and tinnitus, with women being more strongly affected by the distress than men. Researching on the relationship between insomnia and tinnitus, Crönlein and colleagues concluded that tinnitus-related distress is linked to insomnia [13]. These results were confirmed in a Dutch study as specific factors related to tinnitus, such as cervical pain, benzodiazepine and antidepressant usage, and especially an increased intensity of tinnitus was suggested to impair the sleep quality [9]. Psychological models such as the cognitive-behavioural model and the fear-avoidance model provide explanations for the “degree” of suffering caused by tinnitus. Both approaches consider that cognitive processes play a primary role in the experience of tinnitus as well as in clinical management. Central to the cognitive-behavioural model of tinnitus is the high levels of arousal or stress triggered by negative thoughts, which consequently induce autonomic arousal and emotional distress [14]. Cima et al. suggested that anxiety is associated with greater attention to tinnitus, which is a key component in the maintenance of tinnitus distress. Further progression of the model could provide the psychological perspective with a stronger scientific basis and the development of a more effective therapeutic approach [15]. The study of Seydel and colleagues [16] included 607 female and 573 male patients suffering from tinnitus for longer than 3 months. The analysis of pretherapeutic scores of tinnitus annoyance, perceived stress, proactive coping strategies, sense of coherence, and personal resources and the degree of hearing loss as well as tinnitus pitch and loudness were analysed, and results indicated that female subjects were more annoyed by the tinnitus and perceived more stress in contrast to men. Women also showed fewer proactive coping strategies and less sense of coherence and fewer personal resources, but they had lower levels of hearing loss and tinnitus loudness than males. Especially elderly women (≥60 years of age) had more sleep difficulties compared to male individuals with tinnitus. The perception of subjective suffering regarding sleep pain and depression was also reported to be higher in females compared to male individuals [12,16]. According to the hyperarousal model, insomnia is characterized by significant hyperarousal on an autonomous and central nervous level which is demonstrated by significantly elevated spectral power values in the EEG beta and sigma frequency band during NREM stage 2 sleep [17]. On a structural level, it was found that female tinnitus patients differ from male tinnitus patients in the structures of the orbitofrontal cortex (OFC) extending to the frontopolar cortex in beta1 and beta2 waves [18]. The OFC has been implicated as a key region involved in the emotional processing of sounds by previous studies [19,20]. Building on this notion, data from a Belgian study conducted by Vanneste and colleagues showed that the functional alpha connectivity is increased between the OFC, insula, subgenual anterior cingulate (sgACC), parahippocampal (PHC) areas and the auditory cortex in females. They suggested increased functional connectivity in female individuals that binds tinnitus-related auditory cortex activity to auditory emotion-related areas via connections between the PHC areas and the sgACC [18]. In summary, a growing body of literature suggests that neurophysiological items such as enhanced beta activity can be found in a chronic tinnitus and insomnia population, which also has implications on the experience of nonrestorative sleep.

**Study Goals:** We conducted a retrospective and explorative data analysis for items differences in sample 1: all patients with tinnitus and sample 2: patients with tinnitus and insomnia. We searched for differences in the summed test scores and on item-level of the validated psychometric instruments and for gender differences in the tinnitus and sleep anamnesis.

## 2. Materials and Methods

### 2.1. Study Participants and Procedure

All included subjects were patients of the outpatient clinic for Sleep Disorders and Tinnitus of the Department of Psychiatry and Psychotherapy between 2014–2018. Insomnia was diagnosed according to the ICSD-3 criteria, and tinnitus was diagnosed in the outpatient clinic for ORL (Otorhinolaryngology) according to ICD-10 for tinnitus diagnosis [21]. All patients gave written informed consent for data collection and the study was approved by the Internal Review Board of the Paracelsus Medical University in Nuremberg, Germany. Inclusion criterion was subjective chronic tinnitus according to the German S3 Guidelines [22] and age >18 Y. Exclusion criteria were objective tinnitus (with a treatable cause), acute suicidality, psychomotoric agitation, presence of unstable psychiatric comorbidities or unstable medical conditions. 

All patients underwent a psychiatric exploration with subsequent anamnesis by doctors and a psychological examination by psychologists in the out-clinic department for psychiatric sleep medicine and tinnitus according to the diagnostic criteria of ICD-10. The demographic data include educational background, living condition, weight, and status of employment. We applied several psychometric tests using validated questionnaires: MADRS (Montgomery Asberg Depression Rating Scale) [23], BSI (Brief Symptom Inventory) [24], ESS (Epworth Sleepiness Scale) [25], and WHO-5 (5-item World Health Organization Well-Being Index) [26]. 

Tinnitus was evaluated with the TF (Tinnitus Fragebogen)-German Version [27]. The TQ (Tinnitus Questionnaire English Version) total score is the summation of 40 items with 2 items counted double. The different subscales are emotional distress, cognitive distress, sleep disturbance, auditory perceptual difficulties, somatic complaints, and intrusiveness. According to the recommendations of the Tinnitus Research Initiative [28] we included the BDI-II (Beck Depression Inventory) [29] and TSCHQ (Tinnitus Sample Case History Questionnaire) [30].

The anamnesis of insomnia and tinnitus was based on standardized structured interviews, asking questions on sleep latency, sleep duration, frequently awakening, daily functionality, tiredness, hyperacusis, dysacusis, use of a noiser, use of a hearing aid device, tinnitus-triggering movements of the neck and jaw, headaches, or stress. All participants were asked to describe the subjective loudness and the percentage of awareness of the tinnitus using a visual scale between 0 and 100. Furthermore, they had to describe the type and the frequency of their tinnitus. The data sample consists of 33 middle-aged women (mean age: 54.9 ± 12.7) and 43 men (mean age: 50.4 ± 11.2), of whom 23 women (mean age: 56.2 ± 12.1) and 26 men (mean age: 51.2 ± 16.0) have additionally been diagnosed with insomnia using the ICSD-3 Classification of sleep disorders [31].

### 2.2. Statistical Methods

Data were carefully checked for consistency. The actual process of data cleaning involved removing typographical errors, validating, and correcting values against a known list of entities to obtain valid data. Pearson’s Chi-Square, Maximum-Likelihood and Fisher’s Exact test were used to analyse cross tabulations. Independent Student t-tests were used to test parametric and Mann–Whitney U tests were used for nonparametric analysis of continuously distributed variables. All tests were two-sided, and *p*-values < 0.05 were considered statistically significant. All statistical analyses in this report were performed by use of STATISTICA 13 (TIBCO Software Inc., Palo Alto, CA, USA) [32].

## 3. Results

### 3.1. Sociodemographic Characteristics

As shown in Table 1, most patients were married (73.3% of women; 60.0% of men). More women than men did not graduate beyond an intermediate school-leaving certificate (93.3% vs. 69.2%; *p* < 0.03). Most of the patients had completed a vocational training (93.3% of women; 52.0% of men), and only a minority held a university degree (0.0% of women; 24.0% of men). Most participants were employed during treatment (full time: 40.0% of women; 48.0% of men; part time: 33.3% of women; 12.0% of men). As seen in Table 2, the sociodemographic data of the subsample, patients diagnosed with both tinnitus and insomnia are consistent with the data of the entire sample (Table 1).

### 3.2. Psychometric Results

One-third (*n*= 33; 45.2%) of the whole sample suffered from headache, 29 patients (39.7%) from dizziness, 16 (21.9%) from jaw problems and 47 (63.5%) from neck pain. Headache and neck pain were more frequently reported by women (65.6% in women vs. 30.2% in men; *p* < 0.003; 81.8% in women vs. 51.2% in men; *p* < 0.006). Female Patients reported that loud sounds in the surroundings induce pain and discomfort (*p* < 0.03). Women also reported more frequently reported issues with locating sounds due to their tinnitus (61.5% vs. 100.0%; *p* < 0.02) and stated that sounds in the environment were able to mask their tinnitus temporarily (59.4% vs. 43.2%; *p* < 0.03). Women reported much more commonly an increase in tinnitus loudness under stress than men (90.3% vs. 68.2%; *p* < 0.03). On a self-report scale from 1 to 6 for nervousness and restlessness, with higher values corresponding to more nervousness, more women scored 5 or 6 (50.0% vs. 4.8%; *p* < 0.02) and more women reported bad appetite (*p* < 0.04). In the WHO-5, more women than men reported finding everyday life to be filled with things that interest them at least most of the time (33.3% vs. 0.0%; *p* < 0.04). In the MADRS, more women than men reported normal, as opposed to reduced, levels of interest for their surroundings and other people (56.3% vs. 18.5%; *p* < 0.04). In the BSI, more women than men reported feeling at least somewhat guilty (42.9% vs. 10.0%; *p* < 0.02).

As depicted in Figure 1, on aggregate test scores, the MADRS showed gender differences, with men scoring higher on average (22.3 points) than women (16.4 points; *p* < 0.05). The following significant differences, also shown in Figure 1, were found on item level: Men reported a sudden beginning of their tinnitus in 50% as opposed to a gradual buildup than women (28.1%; *p* < 0.05). Men reported to feel inactive because of tinnitus (*p* < 0.05), but if they found some interesting activity, they were able to forget the tinnitus for a short period of time (*p* < 0.02). In the TF, 15.8% of all men, but only 2.4% of all women disagreed with the statement, “There is hardly anything to be done to cope with the tinnitus” (*p* < 0.009). In the BDI-II, more men than women reported finding less pleasure in the things they used to enjoy (100.0% vs. 75.0%; *p* < 0.04).

No gender differences could be found in the following confounding factors: laterality, hyperacusis, dizziness and jaw problem. In addition, the WHO score did not show any gender-correlated significant differences.

In the subsample of patients with tinnitus and insomnia, no gender differences on aggregated test scores were found. No gender differences were further found on the severity of depression, neither for the BDI-II (*p* < 0.35) nor for the MADRS (*p* < 0.52). Yet, on item-level, more women disagreed with the statement, “I am more liable to feel low because of the noises” than men (30.4% vs. 4.0%; *p* < 0.05). Also, more women reported suffering from headaches (70.0% vs. 27.3%; *p* < 0.007) and neck pain (85.7% vs. 50.0%; *p* < 0.02) as shown in Figure 2.

In the TF, men were more likely to disagree with the statement, “There is hardly anything to be done to cope with the tinnitus” than women (20.0% vs. 0.0%; *p* < 0.03). In the BDI-II, more men reported finding less pleasure in the things they used to enjoy than women (52.0% vs. 13.6%; *p* < 0.03). In the MADRS, more men than women reported critical levels of difficulty getting started or slowness initiating and performing everyday activities (81.8% vs. 0.0%; *p* < 0.008).

### 3.3. Insomnia and Psychiatric Comorbidities

The prevalence of insomnia was 64% (95% CI: 53–75), and the most common psychiatric diagnoses according the diagnostic criteria of ICD-10 [33] in the whole group and in the subsample were moderate depressive episodes (24.4% of women, 19.7% of men), followed by anxiety disorders (17.1% of women, 6.6% of men), mild depressive episodes (7.1% of women; 9.8% of men), recurrent depressive disorders (7.3% of women; 1.6% of men), and severe depressive episodes (0.0% of women; 3.3% of men). Anxiety disorders were suspected without secure diagnosis in an additional 26.8% of women and 24.6% of men. A minority of the sample were in psychological or psychiatric treatment elsewhere at the start of their treatment with us (21.2% of women; 31.0% of men).

### 3.4. Tinnitus Related Symptoms

A minority of the patients suffered from hyperacusis (23.1% of women; 16.2% of men) or dysacusis (32.1% of women; 32.6% of men). A relevant proportion of participants reported the use of mechanical devices such as noisers (36.4% of women; 38.5% of men) or hearing aids (38.5% of women; 18.0% of men). Regarding the question of laterality of tinnitus, 11 patients (14.7%) reported lateralisation in the right ear, 32 (42.7%) in the left ear, 23 patients (30.7%) did not report any lateralisation, and 9 (12.0%) described fluctuating localisation. Hearing loss was reported by 35 patients (47.9%). The average subjective tinnitus loudness did not differ between genders (54.5 of 100 in women; 58.4 of 100 in men). In the subsample of patients with tinnitus and insomnia, 18% used noisers and 38.5% of all participants used hearing aids. Cross tabulations analyses revealed no significant associations between hearing aid use, insomnia, and severity of depression, nor did the factors interact with gender.

## 4. Discussion

According to the studies of tinnitus and insomnia, psychological and physiological mechanisms seem to be similar in chronic tinnitus and primary insomnia, including dysfunctional beliefs, negative thoughts, and hyperarousal. However, in tinnitus patients, the focus of negative thoughts and emotions is related to both tinnitus and sleep and not to sleep alone [13,34].

In our patient sample, several gender differences were found, indicating significantly higher scores for depression and clinically relevant symptoms of depression in male patients. Men expressed reduced levels of interest for their surroundings and other people. They further found less pleasure in the things they used to enjoy and reported to finding fewer things that spark interest in them in everyday life. Depressive symptoms in male individuals were higher both in self and in external assessment scales.

Men reported a sudden beginning of their tinnitus. Compared to women, they reported less masking of tinnitus by sounds of the environment. Men were, however, more likely to disagree with the statement, “There is hardly anything to be done to cope with the tinnitus”, which may indicate that men have more coping strategies than women. This leaves open the question of whether more coping strategies are associated with higher or lower acceptance of the chronical tinnitus in men. Ultimately, the need for targeted treatment of mood and depressive symptoms in male patients is indicated.

Male patients diagnosed with both insomnia and tinnitus showed, similar to men in the overall sample, more pronounced depressive symptoms regarding the ability to find pleasure in the things they used to enjoy. They further exhibited critical levels of difficulty getting started or slowness initiating and performing everyday activities, while women displayed higher level of helplessness as well as fewer coping strategies.

The impact of stress is higher in women with tinnitus and in the subsample of women with tinnitus and insomnia because women more often reported an increase in tinnitus loudness under stress and stated to suffer more than men from headaches, neck pain, nervousness, restlessness and feeling guilty, which indicates higher severity of psychosomatic symptoms in women.

The severity of tinnitus and risk of insomnia may be also influenced by socioeconomic factors such as education and employment status [35,36,37]. In our samples, male participants had higher education degrees, whereas in the subsample of participants with tinnitus and insomnia, men had not only higher education but also better employment status. Further research is needed to investigate the causal relationship between the socioeconomic variables and tinnitus or insomnia severity in both genders.

## 5. Conclusions

Our results suggest that chronic tinnitus can be linked to associated worries about sleep or negative emotions that are typical for insomnia.

In our study, men showed more depressive symptoms than women. In contrast to men, women suffered more from psychosomatic symptoms and stress-related worsening of tinnitus, indicating the need for a targeted psychotherapeutic approach including stress management.

Considering the reported differences, tailored therapeutic measures could be implemented to improve both the insomnia and tinnitus in women and men.

## Figures and Tables

**Figure 1 ijerph-18-03209-f001:**
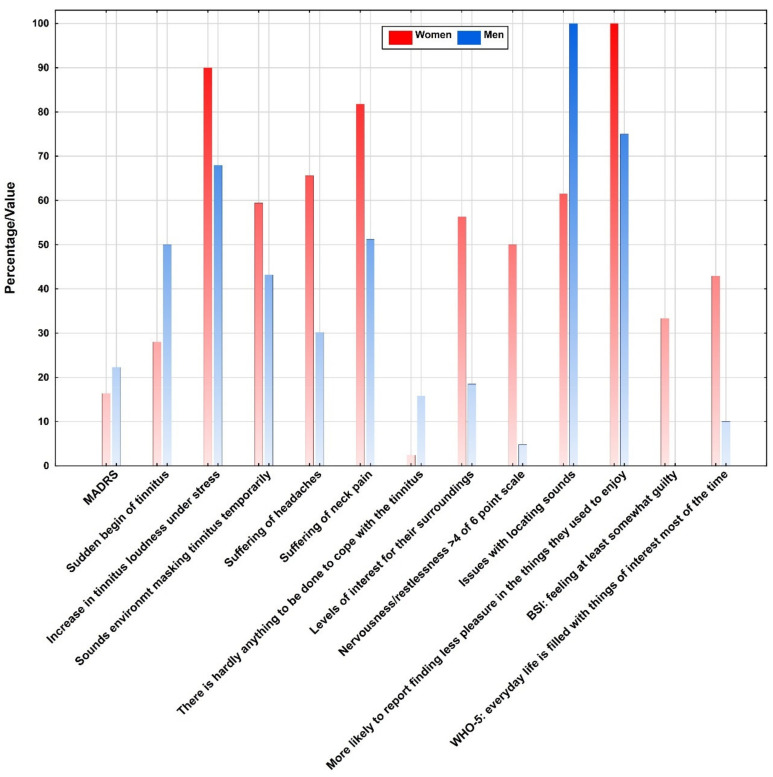
Overall percentage distribution of the statistically significant psychometric outcome variables collected in the total sample.

**Figure 2 ijerph-18-03209-f002:**
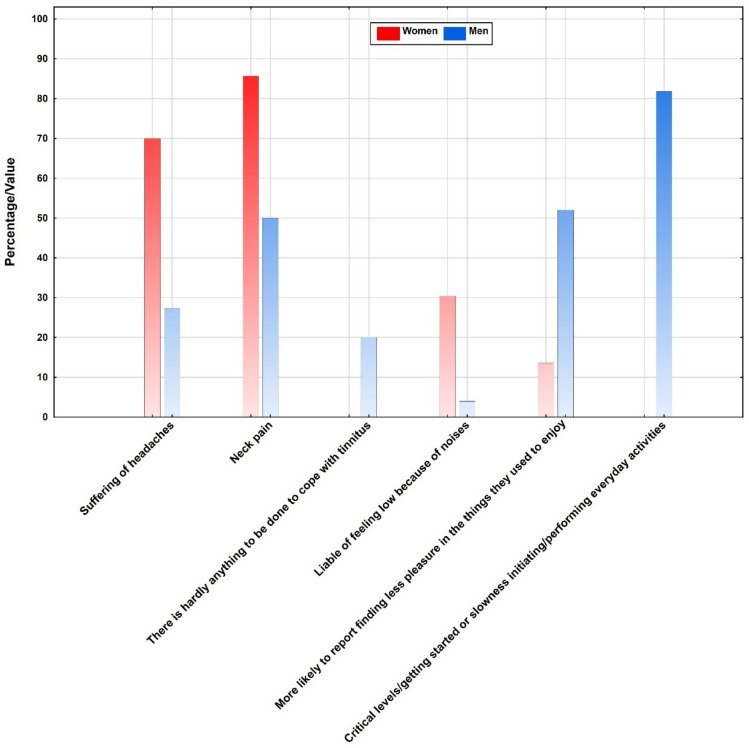
Overall percentage distribution of the statistically significant psychometric outcome variables collected in the subsample.

**Table 1 ijerph-18-03209-t001:** Overview of gender differences regarding sociodemographic data, psychiatric diagnoses, and tinnitus-related symptoms in the entire sample. * significant differences with *p* < 0.05.

Sociodemographic Data	Women (%)	Men (%)	*p*-Value
Married	73.3	60.0	0.22
Higher education (at least intermediate school-leaving certificate)	6.7	30.8	0.02 *
Vocational training	93.3	52.0	0.01 *
University degree	0.0	24.0	0.04 *
Working full time	40.0	48.0	0.48
Working part time	33.3	12.0	0.19
No employment	26.7	40.0	0.43
**Symptoms and diagnoses**			
Year of first reported psychological issue (Ø; year)	2010	2010	0.58
Hyperacusis	23.1	16.2	0.49
Dysacusis	32.1	32.6	0.97
Noiser	36.4	38.5	0.92
Hearing aid	38.5	18.0	0.15
Mild depressive episode	7.1	9.8	0.63
Moderate depressive episode	24.4	19.7	0.62
Severe depressive episode	0.0	3.3	0.24
Recurrent depressive disorder	7.3	1.6	0.16
Anxiety disorder	17.1	6.6	0.24
Suspected anxiety disorder	26.8	24.6	0.24
Tinnitus loudness (Ø; scale from 0 to 100)	54.5	58.4	0.92
Other psychological or psychiatric treatment	21.2	31.0	0.34

**Table 2 ijerph-18-03209-t002:** Overview of gender differences regarding sociodemographic data, psychiatric diagnoses, and related symptoms in the subsample of participants with tinnitus and insomnia. * significant difference with *p* < 0.05.

Sociodemographic Data	Women (%)	Men (%)	*p*-Value
Married	75.0	50.0	0.46
Higher education (at least intermediate school-leaving certificate)	12.5	63.6	0.01 *
Vocational training	87.5	18.2	<0.01 *
University degree	0.0	45.5	0.03*
Working full time	50.0	45.5	0.87
Working part time	50.0	9.1	0.12
No employment	0.0	45.5	0.05
**Symptoms and diagnoses**			
Year of first reported psychological issue (Ø; year)	2009	2012	0.38
Hyperacusis	25.0	9.1	0.17
Dysacusis	20.0	38.5	0.18
Noiser	41.2	33.3	0.84
Hearing aid	31.6	20.8	0.24
Mild depressive episode	14.3	25.0	0.37
Moderate depressive episode	47.6	45.8	0.93
Severe depressive episode	0.0	8.3	0.17
Recurrent depressive disorder	4.8	4.2	0.93
Anxiety disorder	14.3	8.3	0.70
Suspected anxiety disorder	38.1	33.3	0.70
Tinnitus loudness (Ø; scale from 0 to 100)	55.0	62.9	0.68
Other psychological or psychiatric treatment	23.8	38.1	0.32

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
