# Peer review of "Insomnia Associated with Tinnitus and Gender Differences"

_ijerph, 2021, doi:10.3390/ijerph18063209_

Round 1
Reviewer 1 Report
There persists problems of missing figures and supplementary material with the current form of article which needs to be corrected. Apart from this there are some more corrections and comments listed below.
- In general there is stong need to correct the manuscript for typos or grammatical mistakes (e.g., line 26, full stop after restlessness (p<.02),
- Line 42 should read such as rather such.
- Whats consistency and M-L mean in line 122?
- Line 123-124 states as "Independent Student t-tests and Mann Whitney U tests were used to test continuously distributed variables". In my opinion Mann Whitney U tests is used for non-parametric analysis of data and if so, the statement needs to be corrected.
- In Psychometric results section percentage values are not represented uniformly.
- In line 143, p values are not reported in decimal format. This problem persists across the manuscript and needs to be corrected.
- Figure 1 & 2 is missing.
- In table 1 & 2, depiction of % can be omitted from values and simply shown in top section like "women (%)".
Reviewer 2 Report
This retrospective study attempts to characterize the relationship between tinnitus and insomnia. The material consists of 76 tinnitus patients of which 49 (23 females and 26 males) are afflicted by insomnia. Demographic data and an extensive battery of psychometric data are used to characterize the patients. The anamnesis of tinnitus and insomnia was based on standardized structured interviews.
The sociodemographic data of all patients is similar to that of the subgroup of patients with tinnitus and insomnia. More males than females had higher education.
The psychometric tests show that there are gender differences. Females suffer from more psychosomatic symptoms (headache, neck pain, nervousness) and males are more affected by depressive symptoms. Unfortunately my manuscript lacks Fig 1 and Fig 2, which may further strengthen the description of the data.
To summarize, though the material is fairly limited the results are interesting and point toward gender differences regarding tinnitus combined with insomnia. This should be considered when chosing therapy for these disorders. Against this background I recommend this manuscript for publication.
Reviewer 3 Report
The study examines an important symptoms in the natural history of patients suffering from tinnitus. The discussion about gender differences is merited and important in management design. I think the paper suffers from the lack of specialized knowledge about tinnitus in preparation. The introduction can be very much improved by including references to psychological models of tinnitus suffering, like the cognitive behavioural model and the fear avoidance model, to give context to the psychometric evaluation. It was also not clear how tinnitus was evaluated in the clinic and what scales were used to determine severity, and how was that related to the severity of insomnia and depression. It would also be good to look at factors like hearing loss and the use of management protocols (hearing aids, sound therapy), and how did that relate to the severity of insomnia and depression scales, and if those factors interacted with gender
Round 2
Reviewer 1 Report
- Line 135 should read Mann Whitney-U tests were used for non-parametric analysis, right?
- Flipping the axis in figures will make it more presentable but not mandatory.
